# Factors Associated with the Abandonment of Exclusive Breastfeeding before Three Months

**DOI:** 10.3390/children7120298

**Published:** 2020-12-16

**Authors:** Esmeralda Santacruz-Salas, Antonio Segura-Fragoso, Ana Isabel Cobo-Cuenca, Juan Manuel Carmona-Torres, Diana P. Pozuelo-Carrascosa, José Alberto Laredo-Aguilera

**Affiliations:** 1FACSALUD (Facultad de Ciencias de la Salud), University of Castile-La Mancha, Avenida Real Fábrica de la Seda, s/n., Talavera de la Reina, 45600 Toledo, Spain; esmeralda.santacruz@uclm.es (E.S.-S.); Antonio.Segura@uclm.es (A.S.-F.); 2Grupo de Investigación Multidisciplinar en Cuidados (IMCU), UCLM, Avenida Carlos III s/n., 45071 Toledo, Spain; anaisabel.cobo@uclm.es (A.I.C.-C.); Juanmanuel.carmona@uclm.es (J.M.C.-T.); Josealberto.laredo@uclm.es (J.A.L.-A.); 3Departamento de Enfermería, Fisioterapia y Terapia Ocupacional, Universidad de Castilla la Mancha (UCLM), 45071 Toledo, Spain; 4Facultad de Fisioterapia y Enfermería de Toledo, Universidad de Castilla la Mancha, UCLM, Avenida Carlos III s/n., 45071 Toledo, Spain; 5Instituto Maimónides de Investigación Biomédica de Córdoba (IMIBIC), 14004 Córdoba, Spain

**Keywords:** breastfeeding, exclusive breastfeeding, prevalence, Spain, predictors, breastfeeding initiation, supplementary feeding, abandonment of breastfeeding

## Abstract

The commencement and maintenance of exclusive breastfeeding (EB) are dependent on several mother–infant factors. To analyse the prevalence of EB during four different periods and to analyse the factors that can affect its maintenance, we performed a prospective, observational, longitudinal study of 236 mothers and their child between 37 and 42 weeks of gestation and weighing more than 2.5 kg. Four interviews were conducted (T1: on discharge, T2: at 15 days, T3: at one month, T4: at three months). The results showed that EB decreased considerably at three months (69.5% vs. 47.46%). The factors that reduced the risk of abandonment were as following: having decided before giving birth that one wants to offer breastfeeding (T2: odds ratio (OR): 0.02, *p* = 0.001), T3 (OR: 0.04, *p* = 0.001) and T4 (OR: 0.07, *p* = 0.01)) and having previous experience with EB (T2 (OR: 0.36, *p* = 0.01), T3 (OR: 0.42 *p* = 0.02) and T4 (OR: 0.51, *p* = 0.03)). The factors that increased the risk of EB abandonment were offering feeding bottles in the hospital (T2 (OR: 11. 06, *p* = 0.001); T3 (OR: 5.51, *p* = 0.001) and T4 (OR: 4.43, *p* = 0.001)) and thinking that the infant is not satisfied (T2 (OR: 2.39, *p* = 0.01) and T3 (OR: 2.34, *p* = 0.01)). In conclusion, the abandonment of EB in the first three months is associated with sociodemographic and clinical variables and psychological factors such as insecurity and doubts of the mother during the process and the absence of a favourable close environment.

## 1. Introduction

The importance of the practice of maternal breastfeeding is widely recognised throughout the world. Institutions such as the World Health Organisation (WHO) and the United Nations International Children’s Emergency Fund (UNICEF) recommend that it should be exclusive until the infant is six months old and complemented by another type of food until the infant reaches two years of age. These recommendations are based mainly on the benefits it brings regarding child health and development [1,2,3]. 

For almost 30 years, there have been different international initiatives in this field, such as the Innocenti Declaration [4], which established regulations such as the International Code for the Marketing of Breast-Milk Substitutes [5] and the implementation of the Ten Steps Towards Successful Breastfeeding [6]. However, the incidence of maternal breastfeeding continues to be low and does not reach the objectives that have been set. The worldwide level to be reached by 2025 is a minimum of 50% exclusive breastfeeding (EB) at six months [7]. In August 2017, the WHO, the UNICEF and the whole of the Global Breastfeeding Collective evaluated the percentage rates of breastfeeding in 2017 in 194 countries [8]. They found that there was not a single country in the world that observed the policies and recommendations relating to maternal breastfeeding. They also identified that, worldwide, only 40% of infants under six months of age were breastfed by their mothers and only 23 countries recorded such percentages of more than 60%. The percentages oscillated from 24.9% breastfed at six months in countries with medium–high incomes to 43% in countries with medium incomes and even to as much as 64% in Southern Asia (countries with low incomes) [9]. In Spain, according to the latest National Survey of Health in 2017, 39% of six-month-old infants were breastfed, compared with 15.1% in 1995 [10]. 

The critical time points for the early abandonment of EB have been described in the bibliography as the time point of discharge from hospital and the last receipt of maternity benefit in the case of women who receive unemployment benefit [11,12]. According to a recent study, approximately half of the mothers who offer breastfeeding initially do not fulfil their expectations for one month [13]. These authors report that the most critical time point for abandonment is after 12 days of the infant’s life, and this agrees with the findings of other authors [12,13]. 

Several authors have used a theoretical approach to clarify the factors that interfere with the practice of breastfeeding, which could be reasons for its early abandonment [11,14,15]. This theoretical basis explains abandonment as a result of interacting factors as following: (i) the individual level (biological and personal history of the person), (ii) the level of relationships and the community and (iii) the social and cultural level [15]. The model helps to distinguish all the variables that can potentially influence a woman’s decision to abandon maternal breastfeeding prematurely, and it helps to understand how those levels interact among themselves. There is no doubt that the factors that can interfere with the practice of breastfeeding are many and complex and include clinical factors such as psychological factors, state of mind, false beliefs (such as the feeling of hypogalactia), variables in the social environment and other factors related to the mother’s immediate surroundings, as well as cultural factors [14,16,17]. 

The help that mothers should receive about breastfeeding, for both initiation and maintenance, must also be accompanied by practical support. This can vary considerably among the different centres and/or healthcare workers and depends on the idiosyncrasies and characteristics of each community. For its part, the international Baby-Friendly Hospital Initiative (BFHI), proposed by the WHO and the UNICEF, aims to unify assistance with childbirth and breastfeeding [18,19]. In all the centres in which it has been implemented since 1991, it has managed to increase the rate of maternal breastfeeding and improve the health of the community [20,21]. The evidence shows that these benefits extend from improvements in the state of health of the mother and child to benefits for the environment and the economy and thus for the community at large [22,23,24].

In this study, we aimed to identify the most crucial periods for the abandonment of EB before the mother starts or returns to work. In Spain, every woman in paid employment is entitled to maternity leave, paid at 100% for 16 weeks following childbirth [25,26]. In addition, we had the following secondary aims: (a) to identify the prevalence of the different ways of feeding milk upon discharge from hospital, at 15 days, at one month and at three months in a healthcare area which does not yet have BFHI accreditation; and (b) to determine the factors that have the strongest influence on early abandonment at this stage, which is highly significant in the process and practice of breastfeeding.

## 2. Materials and Methods

### 2.1. Subjects and Study Design

The study was observational, longitudinal and prospective. Data on Spanish women who had just given birth in a public hospital without BFHI accreditation were gathered consecutively [27]. In Spain, the healthcare services include regular checks on newborn babies. These checks are nationwide and free, starting 15 days after birth.

It was considered that a sample size of 236 women would be needed to estimate a proportion of EB taking place at six months (20%), with a degree of accuracy of ±5% and a confidence interval (CI) of 95%. The criteria for inclusion in the study were as following: mothers who had recently given birth at full term (gestation of 37–42 weeks) to healthy babies who did not need hospitalisation, had not been diagnosed with any illness or health problems and weighed more than 2.5 kg. The following were excluded from the sample: newborns who tested positive during neonatal screening, mothers who had a health-related problem after childbirth, mothers with multiple gestation, those who refused to continue participating in the study and those who decided to have the neonatal checks performed by a private healthcare service.

The sample initially collected consisted of 286 women and their newborn babies. Finally, 50 newborn babies were excluded; see Figure 1.

All the mothers included in the study were informed of the aims of the investigation, and all offered their prior signed consent. The study had received approval from the Ethics and Legislation Committee of the healthcare area concerned. (CEITO. number: 74. Date: 6/06/2014).

### 2.2. Procedure

Data for the study were collected at four different time points: 1, postpartum; 2, at 15 days; 3, at 1 month; and 4: at three months. At time points 1, 2 and 3, data about the way of feeding milk were collected from the newborn babies’ clinical histories. In addition, the data from time point 1, relating only to the three days in hospital, were supplemented by information gleaned from a personal interview with the mother based on an ad hoc questionnaire about sociodemographic and clinical characteristics and about the kind of feeding that had been chosen. The questionnaire was tailor-made by the researchers (Appendix A). At time points 2 and 3, there were interviews with the mothers by telephone, during which the information recorded by healthcare workers in the computerised clinical histories regarding the type of feeding offered to newborn babies was verified. Questions were also asked about other practices connected with breastfeeding, such as the use of a pacifier, the introduction of solid foods or liquids and the reasons for the cessation of EB (Appendix A).

### 2.3. Study Variables

Ways of feeding newborn babies in accordance with the principles of the WHO [28] were shown as following: 

EB (dependent variable) where the only milk consumed is of human origin, taken directly from the breast or bottle oral rehydration solution (ORS), drops and syrups (vitamins, minerals and medications) are allowed when needed.

Mixed breastfeeding (MB) in which lactation is defined as the consumption of human milk along with formula milk.

Formula feeding (FF) which is the consumption of formula milk exclusively.

According to the theoretical model of variables that affect the process of breastfeeding, the classification is as follows [15]:

Sociodemographic variables that are those of individuals, mothers, children or mother–child dyads: age, nationality, educational level, sex and weight of the newborn, marital status, cohabitation as a couple, employment situation, maternity leave, number of previous children, number of days of gestation, anaesthesia and type of delivery.

Variables which are related to the mother’s social and family environment and could in some way influence her attitude and personal beliefs regarding the process of breastfeeding: mother’s knowledge about breastfeeding, gained from previous experience, the length of the periods of EB of previous children, mother’s expectations about the period for which she would like to offer breastfeeding, the partner’s opinion about maternal breastfeeding, the use of a pacifier and continually thinking that her child is still hungry.

Variables related to the hospital environment and the medical staff: taking bottled water/serum and/or artificial milk during the stay in hospital, experiencing difficulties when breastfeeding, reasons given by the mothers for beginning to offer bottle feeding and the person from whom advice is received during breastfeeding.

### 2.4. Statistical Analysis

Basic, univariate descriptive statistical analyses were performed. The qualitative variables were shown by the distribution of percentages in each category. The Kolmogorov–Smirnov test was used to analyse whether the quantitative variables followed a normal distribution pattern, and indicators of central trends (mean or median) and of dispersion (SD or percentile) were given.

Subsequently, an inferential analysis was performed, in which the association between the dependent variable (EB at all periods) and the independent variables compiled were studied using the chi-squared test between two categorical variables and the ANOVA of a given factor to determine the relation between the pattern of feeding milk and the number of days it lasted. A bivariate analysis using odds ratios (ORs) were used to compare the main factors which determine whether EB continued or not. The CI were calculated for a level of confidence of 95%. For the final multivariate logistic regression models, variables showing *p* values of ≤ 0.05 in the bivariate analysis were used. The SPSS statistical package (version 15.0) was used (SPSS, Inc.; Chicago, IL, USA) for the analysis.

## 3. Results

The sample was composed of 236 women with an average age of 32.3 (SD: 5.3) years, and more than half of whom (121, 51.3%) were unemployed. 

Before giving birth, almost all the women in the sample (219, 92.8%) had made decisions to offer their babies breastfeeding. However, the prevalence of the different ways of feeding milk during the initial period and from birth to three months were very different from what the mothers had originally intended to do. Only one woman (0.45%) decided to offer EB for three months and exceeded that period. Of the 77 (35.2%) who wanted to offer it for three to six months, only 11 (14.3%) achieved that. The rest of the women (138, 63%) said that they had not decided on a maximum or a minimum period; 35 (25.4%) of them achieved their objectives. 

It is also noteworthy that, of all the women who begun EB after giving birth, more than half (94, 57.67%) offered their newborn babies supplementary artificial feeding and 58% of all the babies received water or oral serum from bottles.

The data on breastfeeding prevalence are shown in Figure 2.

The classification of the results obtained from the bivariate analyses of the different influencing factors and/or conditions took account of the most critical periods for the abandonment of breastfeeding: after 15 days, after the first month and after three months (Figure 3).

The multivariate analysis showed the variables that had an influence on the maintenance of EB in each of the periods observed. Of all of these, having doubts or thinking that the infant was not getting enough food or especially having offered the infant bottles of artificial milk during the stay in hospital were the factors that had the most influence on the abandonment of EB at 15 days and before the end of the first month. Giving supplements of artificial milk after birth also increased the risk of the abandonment of EB at three months by 4.43-fold. However, there are other factors that protected the maintenance of EB in this first period. If, before the birth, the mother had decided that she wanted to breastfeed her newborn, this reduced the risk of abandonment at 15 days by 98%, at one month by 96% and at three months by 93%. Having previous experience of this practice resulted in 64% avoidance of abandonment at 15 days, 58% at one month and 49% at three months. Newborn infant girls achieved EB for three months 43% more than newborn infant boys. It is also important to note that, of the children who managed to maintain EB for three months, 112 (47.45%) had received an average of two bottle-feeds fewer (SD: 2.9) than those who did not.

The results obtained from the different factors of influence have been classified taking into account the most critical periods for the cessation of breastfeeding, at 15 days, first month and at 3 months (Figure 4).

## 4. Discussion

This study showed that there are some critical time points that influence the abandonment of EB. We also noted that certain conditions in mothers and their babies and some ill-advised practices have an influence on the maintenance of EB during the first three months of the infant’s life. 

In this healthcare area, the prevalence of EB for three months has reached the proposed international objectives and recommendations by Healthy People (40% in 2010 and 46.2% in 2020) [29,30,31]. In recent years, efforts and initiatives to promote and improve breastfeeding have improved its use during the first months of a baby’s life. However, this was not the case of the objective proposed for supplementation with formula in the first 48 h. For 2020, the prevalence of EB for three months was not supposed to exceed 14.2% of children receiving this supplementation, but in our study, it exceeded 50%. As has been demonstrated in other healthcare areas, recommendations about the institutionalisation and revitalisation of practices that are beneficial for the infant, including the implementation of the BFHI initiative, have enabled the rates of increase and improvements to be seen in the community at several levels [18,19,20,21]. For this reason, in view of the effect of their strategies and in accordance with one of the lines of action proposed in the 2019 Regional Health Plan, this healthcare area has begun to work towards achieving BFHI accreditation in the coming years [32].

### 4.1. Critical Periods for Abandonment

In an environment such as this community in Spain, where neither hospitals nor primary healthcare centres assigned to the area have BFHI accreditation, it is important to be aware of the mother/infant, social and environmental conditions that can help to predict the early abandonment of EB. Some authors have indicated two periods, in which the highest rate of EB abandonment takes place in the first months [12,13,33]. In our study, we identified three time points, at which the circumstances differ. The first period, in which there is the highest risk of abandonment, runs from the discharge from hospital to the 15th day of the newborn infant’s life [13]. At this time, many of the problems associated with the management of breastfeeding and of the newborn appear. Again, in this period, the regulated and procedural care and support for breastfeeding, provided by the healthcare system, is lacking, until the mother and the infant are assessed in primary healthcare consultations. In the second period, lasting until one month from birth, individual factors and the feeling of assurance in the management of breastfeeding continue, and there is also help and advice from outside. In the third period, which lasts until three months from birth, other factors, such as the sex of the infant and the mother’s assurance and external support, can influence the decision to abandon EB or not. Other authors reported the mother starting or returning to employment as an important factor in this period [12,23]. However, there is evidence that, very often, starting or restarting employment is not the main reason for not beginning to breastfeed or for the early abandonment of breastfeeding [34,35]. In our study, we analysed the factors that had an influence on the abandonment of EB between the beginning of the stay in hospital and three months after birth. In this period, in Spain, women enjoy maternity leave, so employment is not a factor that influences mothers to abandon EB or not. Even so, mothers who offer EB may well anticipate these changes by using food supplements during the preceding weeks [36]. 

### 4.2. Regarding the Mother’s Circumstances and the Relationship with the Hospital Setting 

During the stay in hospital after birth, the mother is exposed to many stimuli, new situations, and doubts; when breastfeeding, she may have some problems that she does not know how to resolve herself. Previous experience and personal characteristics, at this stage, interconnect with the influences from the mother’s present environment. If the mother’s social and/or family environment is not positive and does not facilitate this particular method of feeding, in the end, it will mask the best intentions of the mother and artificial feeding supplements will appear [37]. 

Today, with the decline in breastfeeding as an aspect of social culture, advice from healthcare personnel about this method of feeding, not only before birth but also during and after it, can be decisive regarding the mother’s choice of the feeding method by which she is going to offer her child. Some studies provided evidence that many mothers do not offer breastfeeding even though they have received information about it [6,7,15]. In the present study, 70% of the mothers who had received advice about EB from healthcare personnel and/or breastfeeding support groups succeeded in maintaining EB during the first month, while 27% of the mothers maintained EB when no professional advice was received and the mother gathered information herself. Possibly, skills and tools that these professionals can bring to the mother are aimed at solving doubts, anticipating problems and providing the security that the mother needs. In addition, this targeted support could also help to control psychological and mood states that are detrimental to the maintenance of EB [16]. 

### 4.3. Regarding the Circumstances of the Mother and of the Newborn and Their Relationship with the Environment

The mother’s social and family environment also have a decisive influence on whether breastfeeding is practised, since it can predispose the mother towards that practice. The mother’s aptitude and prior knowledge are found to be significant in the abandonment or maintenance of EB before three months. Mothers who have previous experience of offering EB are convinced and have decided before giving birth that they want to offer breastfeeding to their babies. These circumstances help to reduce the risk of abandonment before three months. According to our study and as also pointed out by other authors, most women decide before giving birth on the type of feeding that they wish to offer their babies [37]. Other authors also corroborate that the main factor that serves to predict the launch into breastfeeding is a prenatal decision to feed the infant in this way [29,38]. In our study, mothers without a partner managed to maintain EB (57% at 15 days and 51% at one month) for longer periods than those who did have a partner and lived with him. The reason for this could be that, in fact, the most important thing for the mother is to have decided previously and to really want to offer the infant breastfeeding, not allowing anyone to change her conviction. Other studies stress the importance of a partner’s support and collaboration, but not of co-habitation, in the maintenance of breastfeeding [29,39,40]. However, among women who have doubts and are unsure about whether their babies are being correctly fed or whether the quantity received satisfies them, the risk of abandonment of EB at 15 days and during the first month is more than doubled. This reason, which is called “subjective hypogalactia”, is described in many studies as decisive in the abandonment of EB [13,34,35,41]. 

When the family and/or hospital environment is not conducive to successful breastfeeding and sustains the same insecurity as the mother in relation to the correct way to feed the newborn, this can result in offers of or requests for supplementation with artificial milk in the hospital [11,34]. Replacing intake of the mother’s milk or supplementing it with artificial milk would be, as asserted by other authors, a strong reason for the cessation maternal breastfeeding [11,29,42,43]. According to our study, supplementing with artificial milk is the most important factor affecting the abandonment of EB at 15 days (11 times greater risk), at one month (5 times greater risk) and at three months (4 times greater risk). However, when the mother had difficulty when beginning to breastfeed after birth but managed to overcome it, this resulted in a 60% probability that abandonment in the first 15 days would be avoided. These results confirmed that the support and help provided by medical staff are decisive in continuing with EB during these first days. The newborn infant’s first visit for a check-up is at 15 days after birth, and support from primary healthcare personnel is provided, including help with the maintenance of breastfeeding. 

Another factor which, although little studied, is important to be aware of, is the way in which the sex of the newborn infant has an influence on whether EB continues or not. We agree with the result of other study that mothers whose newborn babies are girls are less likely to abandon EB than mothers whose babies are boys [36]. In this study, the former abandoned less in the first three months, and in our study, they abandoned less in the first six months (OR: 3.6, *p* ˂ 0.001). In neither study did we find any reasonable explanation for this. However, we believe that this is associated with the energy demands for growth and development, which are higher in boys than in girls. This generates an increase in the demands of breastfeeding for the mother and results in greater exhaustion. We consider that more research into this question will be needed in the future. 

We also believe that it is important to point out that although the variable “type of delivery” was not included in the multifactorial predictive model, vaginal deliveries with or without instruments, as opposed to caesareans, are associated with greater maintenance of EB up to three months. 

### 4.4. Strengths and Limitations

One of the strengths of this study is that it was a longitudinal investigation using information obtained directly from mothers by means of a direct survey, with responses collected by personnel experienced in this field, as well as by telephone. Moreover, the people entrusted with the fieldwork were not the same as those who analysed the data or those directly involved with the breastfeeding process in the healthcare centres. On the other hand, the study had its limitations; there could have been recall bias when trying to record the precise dates of abandonment of EB. However, the data about each woman were checked both by telephone and in her clinical records. In addition, the characteristics of the questionnaire, prepared ad hoc and used for collecting information, could restrict the observation of other variables that influence abandonment and the comparison with other studies. Due to these limitations and large ORs, future studies need to be conducted in other countries and with larger samples in order to understand in depth the factors associated with the abandonment of exclusive breastfeeding.

## 5. Conclusions

The results of this work confirmed that, in this population, the rates recommended in the Healthy People 2025 EB objectives are reached at three months. However, there are incorrect practices, carried out in the hospital, that interfere with the breastfeeding process.

To offer EB, several factors of the mother and the newborn must act together. Sociodemographic, clinical and environment-related variables are involved. Although there are different predictive conditions of abandonment in each of the three critical periods, at 15 days, at one month and at three months, some are decisive and recurrent in all of them. The most important is the fact that they have been offered bottles of artificial milk during the hospital stay. On the other hand, having decided on the type of feeding to be offered to the infant prior to delivery and having previous experience in offering breastfeeding helps to maintain EB up to three months. However, the help and advice of professional medical personnel is crucial to avoid the cessation of EB, especially in the early days. The support and help of health professionals can avoid difficulties and solve problems and doubts in the mother, including insecurity in the amount of milk produced and therefore the correct feeding of her child.

## Figures and Tables

**Figure 1 children-07-00298-f001:**
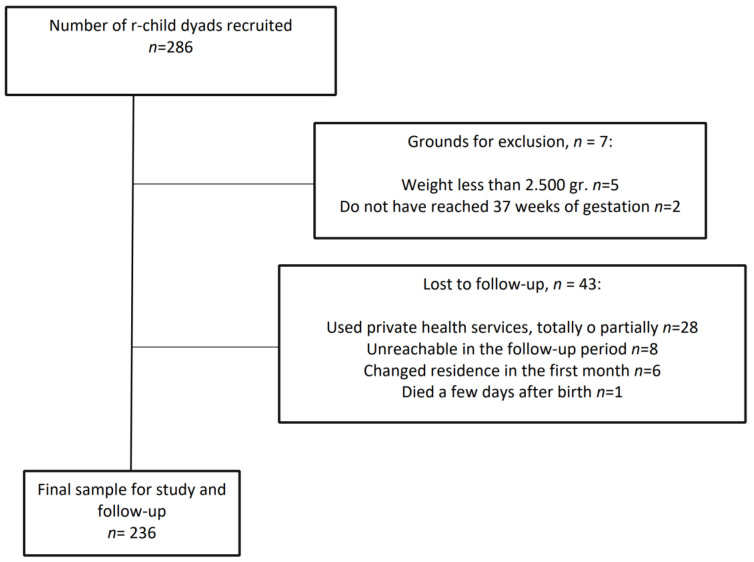
Flow chart of the analysis of breastfeeding practice of the participants in the study during the first three months after giving birth.

**Figure 2 children-07-00298-f002:**
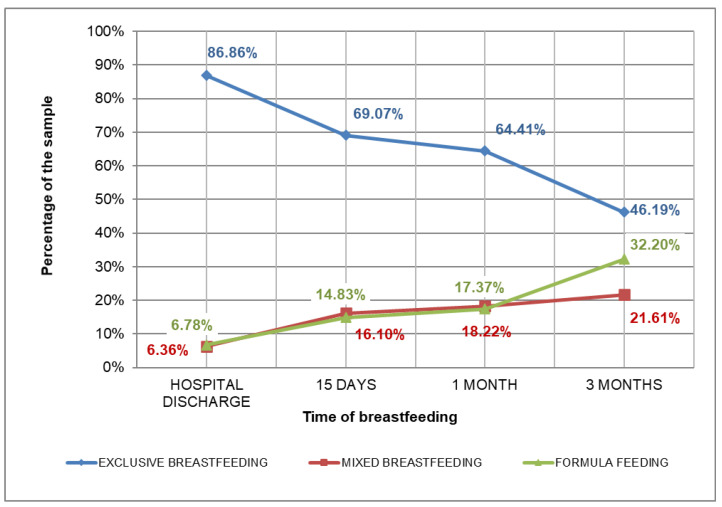
Prevalence of breastfeeding types over a period of three months.

**Figure 3 children-07-00298-f003:**
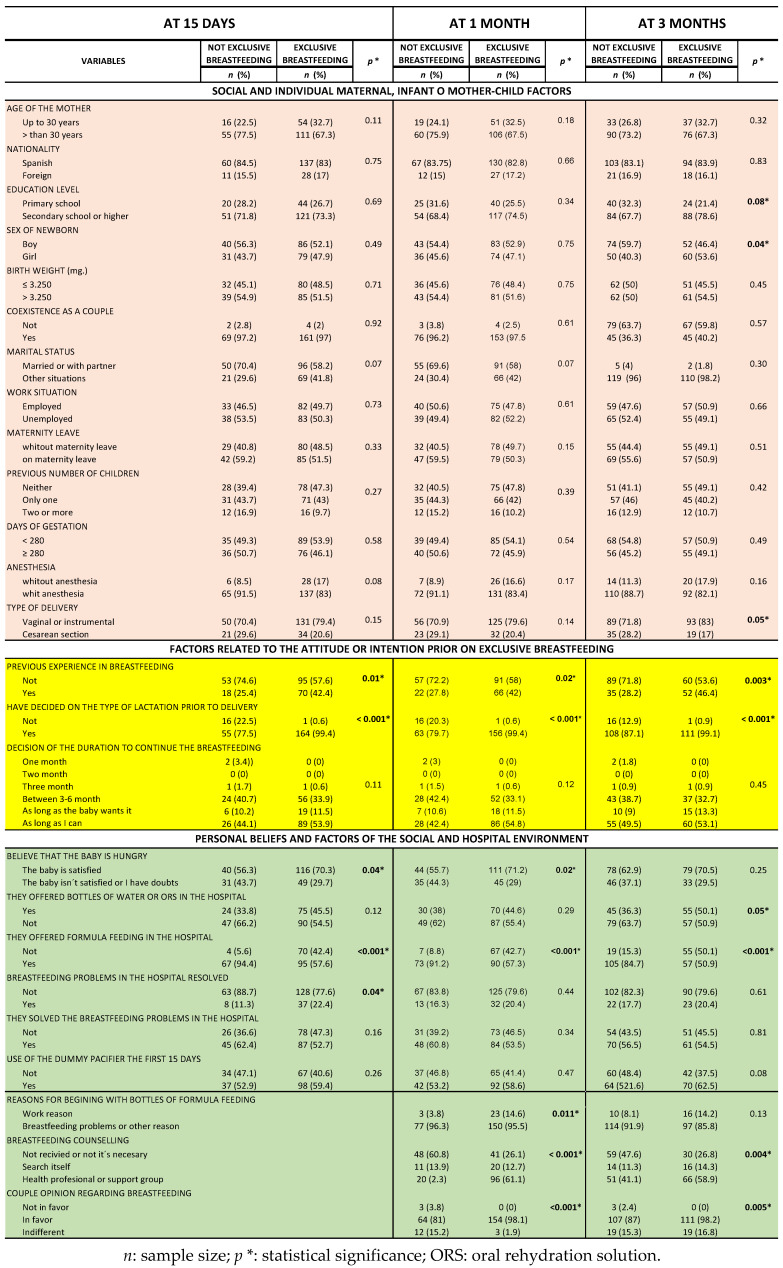
Conditions influencing the maintenance of exclusive breastfeeding up to three months.

**Figure 4 children-07-00298-f004:**
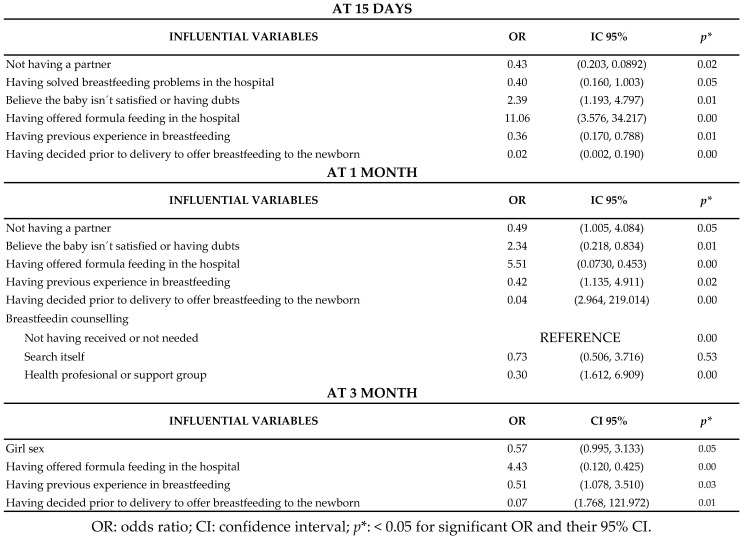
Multivariate predictive models of exclusive breastfeeding abandonment in the first three months.

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
