# Peer review of "Factors Associated with the Abandonment of Exclusive Breastfeeding before Three Months"

_children, 2020, doi:10.3390/children7120298_

Round 1

Reviewer 1 Report

General Observation: 

The title of the manuscript is of interest, captivates Reader's attention. However the Prior and Initial is a bit confusing

Sections of the manuscript written in perfect English while other sections are not. 

Minor changes

Some mix in referencing styles (line 66, 291, 249-250)

Not sure if IHAN is same as IBFAN (Line 84; May be in Spanish?)

Line 107-Use of terms like metabolopathies may be confusing to readers

Line 214- suggest rephrasing for clarity

Line 225-Typo error, suggest correction

Line 233- Unclear, suggest, rephrasing

Figure 2-suggest reduce the two decimal points on percentage (vertical axis)

The use of Ad hoc questionnaire as mentioned in the study Limitation section  is confusing. Please explain explicitly in the methodology section.

Major change

Conclusion: Mixed up, unclear, does not relate to the study findings and too long. This section must be re-written.

Recommendation: seems to be missing or lost in the conclusion. This needs to be written 

Author Response

Factors associated with the abandonment of exclusive breastfeeding before three months.

Response letter to reviewers 

Dear Carrie Wang, Assistant Editor, and anonymous reviewers.

The authors would like to thank you for giving us the opportunity to revise and improve our manuscript; we also thank the reviewers for their thoughtful and constructive comments. We have considered the suggestions and have incorporated them into the revised manuscript. We believe that our manuscript is stronger as result of these modifications. Changes in the original manuscript are highlighted in yellow. An itemized point-by-point response to the reviewers’ comments is presented below :

Kind regards

The authors.

*************************************

Reviewer 2 Report

The breastfeeding pattern is modulated for many variables which included previous maternal concerns, psychological factors, pregnancy or infant health even social supporting. The deep knowledge in these factors report us a powerful beneficial to nurse and also to improve protocols which support the old and new mothers in lactation stage. That not only impact on maternal health but also on infant health and, in long term, in the adults because breastmilk is a gold standard nutrition in newborns. The Santacruz-Salas et al article explores some of these parameters, although are already known, is pretty great to re-new data and also known others strategies in other health care centers. However, some details and recommendations need to be reported as a scientific communication in order to this article would be truly valuable.

Abstract: The conclusion was lazy and poorly focus.

  • Subheadings need to be removed it.
  • Line 25: normal weight and gestational age should be more specify.

Keywords: I suggest if the authors could include the MeSH terminology. “Influencing factors” or “feeding of the newborn baby” really match with the article?

Introduction:  in general was detailed and the background/context presented. However, the authors mentioned some factors which could impact on exclusively breastfeeding cessation, such as psychological factors. I kindly suggest if the authors could review the article (PMID: 33007816) which is related to and how could integrate with their work.

  • Line 72: what the authors means with individual level?
  • Lines 80-81. It is not clear what the authors want to say
  • Acronyms of IHAN need to be defined

Material and methods: I am glad for reporting of sample size calculation and it is interesting which match with the final cohort enrollment. My mayor concern would be that the authors did not control the models by some variables which have been strongly described if related with the breastfeeding patters (i.e.: educational level, economy income, parity, etc…). A strong rationality need to be development, at least in the discussion section.

  • The questionnaire ad-hoc should report in the supplementary material. In addition, there are some index of reliability? How were the ambiguous questions or the socially accepted responses controlled?
  • What is “use of the dummy”?
  • The section “Study variables” need to be strongly re-organized: what the authors means with non-human formula? Work permit would be maternity leave?
  • Statistical analysis: I suggest to separate the descriptive statistical of analytic statistical. In the bivariate analysis, it told about correlations but then, it explained the chi-squared, this test is association test between proportion. How the ORs were used to compare? The comparison should be related to the test or p-value form the model. In the models, what method was used, what IV, what predictors, need to be specified.

Results: it would be interesting report a descriptive table regarding: maternal age, nationality, civil status, employment, gestational age, infant sex, route of delivery, pregnancy complications, educational level, economy income, maternity leave, birth weight and length (or their z-score), Apgar at 5 min, etc… without any comparison, therefore, the models need to be adjusted by these variables which is already know affects the breastfeeding pattern. In addition, in the tables need to report some statistical methods were used.

  • There are many subjective references would not be allowed in this section: line 202 “…showed certain variables…”, line 214 “43% more…”
  • The CI are huge in some cases, could be for the missing values, how were handling?

Discussion: I kindly suggest if the authors could report some references which support more their conclusions. Furthermore, in this article (PMID: 33007816) could find some data which support the EB cessation in the third months related to the maternal psychological factors.

  • They found that infant sex could be a protective or risk factor for EB cessation. Although others authors could not find a rationale, the authors should adventure what hypothesis they propose.
  • Line 281. Would be great if the authors could position with any recommendation.

Conclusion: Although this article is not strongly original, the conclusions are focus and are clear.

Minor comments:

  • “:” before subheadings are not match with the guidelines of the journal. The capitation in the figures should be “figure”. Other type-error need to double-check (i.e.: extra spaces, capital-letters, decimals would be by “.”…).
  • There are many unappropriated terminology (i.e.: “baby” for “infant”; “moment/period” for “time-point”; “history” for “record”; “abandoning” for “cessation”; “manner” for “pattern”). In addition, some construction would not be used in a scientific communication (i.e.: “so”, “don’t”, “Haven’t”)

Author Response

(The authors gave the same response as above.)

Round 2

Reviewer 2 Report

I want to thank the authros the huge effort to report their data and to review the manuscript which I think was strongly imporved. I appreciate the traslocational research with nurse point of view. No more comments from my side.